# Dietary Strategies for Relieving Stress in Pet Dogs and Cats

**DOI:** 10.3390/antiox12030545

**Published:** 2023-02-21

**Authors:** Zhicong Fan, Zhaowei Bian, Hongcan Huang, Tingting Liu, Ruiti Ren, Xiaomin Chen, Xiaohe Zhang, Yingjia Wang, Baichuan Deng, Lingna Zhang

**Affiliations:** Laboratory of Companion Animal Science, Department of Animal Science, South China Agricultural University, Guangzhou 510642, China

**Keywords:** stress response, oxidative stress, dog and cat, dietary strategy, pet food and nutrition

## Abstract

A variety of physical, emotional, and mental factors can induce a stress response in pet dogs and cats. During this process, hypothalamus–pituitary–adrenal (HPA) and sympathetic–adrenal medulla (SAM) axes are activated to produce a series of adaptive short-term reactions to the aversive situations. Meanwhile, oxidative stress is induced where there is an imbalance between the production and scavenging of reactive oxygen species (ROS). Oxidative damage is also incorporated in sustained stress response causing a series of chronic problems, such as cardiovascular and gastrointestinal diseases, immune dysfunction, and development of abnormal behaviors. In this review, the effects and mechanisms of dietary regulation strategies (e.g., antioxidants, anxiolytic agents, and probiotics) on relieving stress in pet dogs and cats are summarized and discussed. We aim to shed light on future studies in the field of pet food and nutrition.

## 1. Introduction

With the improvement of living standards and changes in population structure (i.e., increasing single and geriatric populations), the number of domestic pets has largely increased in recent years, accompanied by the rapid growth of the pet industry and pet-related economy [1]. Meanwhile, welfare concerns for pets have become increasingly prominent [2]. Stressors exist ubiquitously along the pet industry chain, such as exposure to transportation and novel environments, and inappropriate caretaking strategies [3]. Diseases, behavioral problems, and even death can occur in animals if stress is not properly managed [4]. Being part of the stress response, oxidative stress is an important factor in the pathogenesis of many diseases, such as neural dysfunction and inflammatory bowel disease [5], and there is a complicated interaction between oxidative stress and disease progression [6]. In turn, aversive consequences from the stress response can challenge the animal’s welfare, damage the pet–owner relationship, and increase the abandonment of pets, which could exert a threat to public safety [7] and biodiversity [8]. Appropriate management of stress in pets is therefore necessary and urgent.

The current paper attempts to comprehensively describe stress in pet dogs and cats, summarizing its causes, mechanisms, and potential consequences, and mainly focusing on dietary strategies for relieving pet stress. The hypothesis is that dietary ingredients that can address physiological and behavioral changes of stress response may serve as effective modifying strategies for stress management. The aim of this review was to identify the potential of some substances in relieving stress in pet dogs and cats and to provide a reference for the development of new functional pet foods targeting stress management.

## 2. Causes of Stress in Pets

Causes of stress (i.e., stressors) can be classified as physical (e.g., infection, hemorrhage) or psychological (e.g., restraint and threat). Stress in pets rises most commonly in situations when predictability is lacking or when the animal’s needs are not met.

### 2.1. Environmental Factors

Uncomfortable environments can cause chronic stress in dogs and cats. Extreme temperature may lead to cold and heat stress [9]. At the same time, interruption of daily routines [10,11], vet visits [12], and novel environments [13] can also cause stress and anxiety, especially in cats. Abrupt environmental accidents such as sudden noise [14] and falling objects [15] usually result in panic and fear in pets. Even some common feeding practices, such as water-softened dry food can present stress in pets [16]. Psychological stress can occur when space allowance for activity and behavioral needs are not met [17,18]. Dogs and cats kept mostly or strictly indoors with little environmental enrichment may not able to fully perform natural behaviors such as playing and hunting, the frustration from which can cause anxiety and depression, and the exhibition of behavioral and physical problems [19,20].

### 2.2. Social Conflicts

Pets living with humans can be exposed to imbalanced power because we are the ones that control their physical and social environment. Good human–pet relationships and the forming of bonds between pets and owners can provide mutual benefits [21,22]. Inappropriate or aversive interactions with pets can result in compromised or even broken relationships and cause additional stress in pets. Examples include the use of punishment [23,24], social deprivation [25,26], and some seemingly normal or intimate owner behaviors such as restraint [27] and forced interaction [4].

Pets may also encounter other inter- or intraspecific social conflicts, such as territorial disputes and miscommunication [28]. Conditions of limited space or resources (e.g., food) further increase the possibility of conflict outbreak [29]. The introduction of a new cat to a stable colony may interrupt the original social dynamic and cause fighting [30,31]. In addition, non-contact aversive social stimuli, such as exposure to dog barking, can also cause stress in cats [28].

## 3. Mechanisms of Stress

Stress response is elicited when an actual or potential threat to the homeostasis of the organism is perceived [32]. The process involves the activation of the hypothalamus–pituitary–adrenal (HPA) and sympathetic–adrenal medulla (SAM) axes as shown in Figure 1 [33,34]. As a result, changes in various physiological processes and behaviors are induced [33,34]. Oxidative damage is also incorporated in sustained stress response causing a series of chronic problems [6].

### 3.1. SAM and HPA Axis

Excitement of the sympathetic nervous system in the SAM axis promotes the release of acetylcholine from preganglionic fiber endings and the postganglionic neurotransmitter noradrenaline, which acts on the adrenal medulla that is located above the kidneys on both sides of the spine in the retroperitoneal space, thereby promoting the release of catecholamine (i.e., adrenaline and noradrenaline) into the bloodstream [35]. Blood redistribution occurs after SAM axis activation, leading to vasoconstriction in many microvascular networks and vasorelaxation in skeletal muscle and liver [36]. This accelerates cardiac contraction, thereby increasing blood output and blood pressure. The SAM axis responds to stress rapidly to get the animal ready for the “flight or fight” reaction.

The HPA axis includes the hypothalamic paraventricular nucleus (PVN), a hollow funnel-like region located inside the supraoptic area of the hypothalamus, the pituitary (an oval body located in the ventral hypothalamus), and the adrenal gland. After exposure to stress stimulation, the PVN secretes corticotropin-releasing hormone (CRH) and arginine vasopressin (AVP) to the portal circulation of the median eminence [37], where CRH quickly reaches the pituitary gland and promotes its secretion of adrenocorticotropic hormone (ACTH). ACTH acts on the adrenal cortex to promote the secretion of glucocorticoids (GCs), such as cortisol and corticosterone. Glucocorticoid secretion negatively regulates CRH and ACTH secretion. The activation of the HPA axis, and the subsequent increased content of GCs, modulates energy reserve mobilization and catabolic processes, such as to promote gluconeogenesis and increase protein and fat metabolism through proteolysis and lipolysis. Meanwhile, certain physiological processes are temporarily inhibited, leading to immune suppression and the inhibition of digestion, reproduction, and growth [38,39]. The activation of the HPA axis is relatively slow [39,40]. Although the negative feedback mechanism will restore it to its normal level, excessive cortisol from long-term chronic stress brings serious health risks to the body [39,40].

### 3.2. Oxidative Stress

Oxidative stress refers to when the production of oxidants exceeds the antioxidative capacity of the body, leading to the disruption of redox (i.e., oxidation/reduction reactions) homeostasis and ubiquitous damage to cellular, tissue, and organ systems [41]. The mechanism of oxidative stress is shown in Figure 2.

Reactive oxygen species (ROS) widely refers to oxygen-derived free radicals and non-free radicals. In normal cellular activities, oxygen in the mitochondrial inner membrane will gain electrons under the action of the respiratory chain and produce ROS with high chemical reactivity due to unpaired electrons [42]. Other main sources of ROS include enzymes such as NADPH enzyme oxidation [42], cytochrome P450 in the endoplasmic reticulum, lipoxygenase, xanthine oxidase, and cyclooxygenase [43].

Meanwhile, the body is endowed with a defensive reducing system to combat ROS, which consists of antioxidant proteins, antioxidant enzymes, and small-molecule antioxidants. Antioxidant proteins include mainly albumin, haptoglobin, ferritin, ceruloplasmin, etc. [44]. Antioxidant enzymes include superoxide dismutase (SOD), catalase (CAT), glutathione peroxidase (GPx), and some coenzymes. Small-molecule antioxidants are divided into lipid-soluble and water-soluble antioxidants [6].

Under normal circumstances, ROS and the antioxidant system maintain a relative balance. However under stressful conditions, ROS will be overproduced, which may lead to oxidative stress [45]. The increased respiratory rate [46], blood glucose [47], and the secretion of glucocorticoids [48] and catecholamines [49] during stress response are all proven to induce ROS production. On one hand, the production of ROS promotes the activities of antioxidant enzymes through the Nrf2-Keap1 pathway [41]. On the other hand, an inflammatory reaction is induced, mainly through the Nf-κB pathway [50].

The level of oxidative stress can be reflected by the typical byproducts from the process of oxidative damage. For example, malondialdehyde, 4-hydroxynonel, F2-Isoprostane, and oxygenated low-density lipoproteins are derived from polyunsaturated fatty acids during lipid peroxidation [6,41]. The sulfhydryl group in protein is also easily attacked by ROS, which is converted to carbonyl protein [51]. Reactive carbon groups, such as advanced glycation end products (AGEs), may be generated during glycosylation of protein under the action of glucose ROS [52]. With nucleic acid, ROS may also attack guanine to generate 8-oxo-2’-deoxyguanosine (8-oxo-dG) in DNA and 8-oxo-guanine (8-oxo-G) in RNA [53]. Reduced glutathione (GSH) and glutathione disulfide (GSSG) are two forms of glutathione that play important roles in protein redox [54]. GSH is oxidized to form GSSG [55], and an increase in ROS usually leads to a loss in GSH; therefore, the ratio of GSH/GSSG also can serve as a measure of oxidative stress [55].

The brain has high oxygen consumption and is rich in lipid content, which makes it vulnerable to oxidative stress [56]. As a result, chronic ROS accumulation presents a threat to the integrity of brain cells and neural functions, disrupting neural circuits, impairing connections between the hippocampus, amygdala and cortex, and ultimately leading to behavioral and cognitive deficits [57].

In addition, oxidative stress can induce inflammatory responses through the NF-κB pathway [50]. Inflammatory responses can induce inflammatory bowel disease (IBD) and affect the kynurenine pathway to worsen neurological and intestinal health, thus aggravating stress or increasing sensitivity to stressors [58]. IBD is a chronic gastrointestinal disease that is usually associated with stress [59]. Although the exact mechanism remains to be explored, more and more studies have shown that oxidative stress plays a crucial role in the pathogenesis and progress of IBD [50]. Oxidative stress caused by excessive ROS may stimulate the initial inflammatory reaction and lead to additional ROS production which may result in further damage to the intestinal tissue [50]. In the intestinal tract, tryptophan can be used to synthesize 5-hydroxytryptamine and kynurenine (KYN); the latter can be further broken down to produce kynurenic acid (KYNA) and neurotoxic quinolinic acid (QUIN) [60,61]. Under inflammatory conditions, increased QUIN synthesis results in the depletion of gamma-aminobutyric acid (GABA) and adenosine triphosphate (ATP), which further aggravates damage to nerve cells [62].

Generally speaking, damage caused by oxidative stress has a negative impact on many tissues and organs, and if it is not alleviated, it may cause or mediate the progression of a series of problems or diseases [63].

## 4. Adverse Consequences of Stress

### 4.1. Gastrointestinal Diseases

Studies have shown that the number of gastroenteritis cases in cats during the SARS-CoV-2 pandemic was higher than in the pre-pandemic period, due to stress as a result of the changes especially influencing the daily routine of cats [64]. Chronic stress may lead to gastrointestinal ulcers (i.e., lesions of the gastric and duodenal mucosa), which manifest as mucosal erosion, bleeding, and even perforation [65]. Secretion of catecholamines during stress response decreases the blood flow to the gastrointestinal system causing mucosal ischemia [66]. Meanwhile, intestinal hypoxia can lead to ATP depletion, acidosis, and the destruction of the gastric mucosal barrier [67]. The H^+^ in the gastric cavity diffuses reversely into the mucosa and further aggravates the gastrointestinal injury. Studies have also shown that glucocorticoids can increase gastrointestinal permeability [68]. Although varied in different stress models and species, it is generally believed that acute stress will lead to the delayed gastric emptying and accelerated transport of the large intestine [69], resulting in diarrhea, vomiting, and other digestive tract problems [70]. The effect of chronic stress on the gastrointestinal tract seems to be sustained even after the stressor is removed. Some studies show that when the stressors are eliminated, the colon still accelerates transport of digesta [69,71], which may be closely related to sustained diarrhea observed in chronic stress. In addition, oxidative stress is considered to be involved in different gastrointestinal diseases in pets, such as feline panleukopenia [72] and inflammatory bowel disease (IBD) in dogs [5].

### 4.2. Immune Dysfunction

Acute stress can enhance innate immunity in order to better cope with adverse changes. The underlying mechanism may be that norepinephrine and other stress hormones induce the recruitment of dendritic cells and the increase in macrophages at antigen exposure sites, thereby enhancing the primary immune response [73]. Under acute stress conditions, the total number of white blood cells also increases [74]. However, continuous activation of the HPA axis will lead to leukopenia [75]. Increased glucocorticoids have been shown to exert a strong immunosuppressive effect by inhibiting cytokine production, macrophage function, lymphocyte proliferation and differentiation, and natural killer cell activity [76,77]. Therefore, chronic stress can lead to immunosuppression and increase the risk of pathogen invasion. For example, contraction of feline infectious peritonitis due to feline coronavirus has been linked to oxidative stress and decreased antioxidant status in cats [78].

### 4.3. Urinary Tract Diseases

Stress can lead to urinary tract problems such as dysuria, hematuria, pollakiuria (i.e., increased frequency of urination), and periuria (i.e., urination in inappropriate locations). On the urethral side, the activation of the renin–angiotensin–aldosterone system and the secretion of catecholamines from the SAM axis lead to renal vasoconstriction and reduced glomerular filtration rate and urine output. Moreover, the increase in the secretion of antidiuretic hormone enhances the reabsorption of water and further reduces urine volume. In the lower urethra, feline idiopathic cystitis is mostly of type I neurogenic origin [79]. It was found that plasma catecholamine concentration at rest in cats with idiopathic cystitis was significantly higher than that in healthy cats [80]. In addition, plasma catecholamine concentrations decreased with stress adaption in healthy cats but remained high in cats with idiopathic cystitis [81]. Collectively, stress can affect urinary tract health through neuroendocrine pathways.

### 4.4. Cardiovascular Problems

The cardiovascular system often reacts to stress with accelerated myocardial contraction and heart rate, and increased blood pressure and cardiac output. The reaction is induced through catecholamines interacting with their β-receptors on myocardial cells [82]. In the long run, the threshold of ventricular fibrillation is reduced due to over-secretion of catecholamines, causing abnormal myocardial activity and arrhythmia [83]. An earlier study in infarcted dogs showed that stressful stimuli provoked diverse ventricular arrhythmias including ventricular tachycardia and early extrasystoles [17]. The more worrying situation with chronic stress is that prolonged secretion of GCs can lead to a permanent increase in cardiac sympathetic tension and hypertension, resulting from elevated blood cholesterol levels and sodium retention in vascular smooth muscle cells [84]. Therefore, high-intensity, high-frequency, or long-term hypertension induced by stress can have adverse effects on the cardiovascular system and even lead to heart disease [84,85]. Meanwhile, oxidative stress seems to be highly correlated with cardiovascular disease. The activity of SOD in cats with hypertrophic cardiomyopathy was decreased significantly [86]. The serum antioxidant capacity of dogs with heart failure also decreased [87].

### 4.5. Acute Stress Behavior and Behavioral Abnormalities

#### 4.5.1. Acute Stress Behavior

When facing acute stress, cats or dogs often exhibit “flight or fight” responses. Cats will try to hide or flee. The typical hiding posture in cats is freezing while squatting and crouching their body [88]. If avoidance of the threat is not achieved, cats will exhibit intimidating and aggressive behaviors, such as hissing, growling, slapping, scratching, and biting [89]. When dogs suffer from acute stress, there will be body shaking, lowering of the posture, mouth licking, and restless walking and standing [15]. They will even show aggressive behaviors, such as barking, lunging, growling, and biting/snapping [90]. Fortunately, mild, transient acute stress does not cause substantial damage to the body. If not alleviated, acute stress may evolve into chronic stress [91], which in turn leads to abnormal behaviors (e.g., stereotypic behavior, urinary marking, aggression).

#### 4.5.2. Behavioral Abnormalities

Stress can cause anorexia nervosa, leading to decreased appetite and food intake in dogs and cats [92]. The neural circuits regulating food intake converge on the paraventricular CRH-releasing nuclei and neurons containing urocortin [93]. CRH exerts an anorexigenic effect by inhibiting the release of neuropeptide Y and other hypothalamic neuropeptides, such as growth-hormone-releasing hormone and somatostatin. The orexigenic effects of glucocorticoids are counteracted by a steroid-induced rise in leptin levels that close a regulatory loop regarding food consumption [94,95]. On the other hand, studies in rats and humans show that stress may also lead to overeating [96,97], an eating disorder involves the brain reward system [98].

Some common obsessive–compulsive behaviors in pets include feline hyperesthesia syndrome, psychogenic alopecia and pica in cats [99], and acral lick dermatitis in dogs [100]. Studies have shown that stress can lead to obsessive–compulsive behavior in dogs and cats, which may be related to the dysfunction of neurotransmitters (e.g., 5-hydroxytryptamine and dopamine). Mami Irimajiri et al. partially confirmed this in dogs and showed that 5-hydroxytryptamine reuptake inhibitors (e.g., clomipramine and fluoxetine) exhibit reliable mitigation effects on obsessive–compulsive disorder [100].

Urine marking is considered a territorial behavior in dogs and cats as urine contains odor information for individual and sex identification [101]. Inappropriate urine marking is especially common in multi-cat households where incompatible or unfamiliar individuals live together [4]. The exhibition of urine marking in response to social conflict is the attempt of cats to gain control of the environment by leaving behind familiar odors [4]. Urine marking is often accompanied by other behavioral problems, such as aggression in cats [24], indicating a close relationship between a general stressful environment and behavioral problems [102].

## 5. Dietary Strategies for Relieving Stress in Pets

Current strategies for relieving stress in cats and dogs commonly include managing their environment, training techniques [103], pheromonotherapy [101,104], and some other olfactory stimuli such as plant-extracted essential oils [105]. Pharmacotherapy may be necessary when the case is severe, but drug administration itself may provoke stress [4]. In recent years, more and more studies have focused on relieving stress through nutritional regulation, which have been mainly focused on effectiveness in anti-oxidation, anti-anxiety, and/or maintaining intestinal health. Studies on the nutritional management of stress in cats and dogs have been summarized in Table 1.

### 5.1. Antioxidants

Exogenous antioxidants are substances that can improve immune function, boost the endogenous antioxidant system, and balance the cellular oxidative status by scavenging free radicals and by interrupting the lipid peroxidation process [140]. The protective role of different natural antioxidants in chronic diseases has been documented in various animal species and humans [140].

#### 5.1.1. Polyphenols and Other Plant Extracts

Antioxidant phytochemicals are commonly found in fruits (e.g., berries, apples, grapes, and pomegranates), cereal grains, vegetables, and plants. The main group is polyphenols, the chemical structure of which contains one or more aromatic rings and can act as free radical scavengers and metal chelators [141].

Gallic acid (GA) is a naturally occurring polyphenol commonly exist in fruits, vegetables, and herbal medicines. GA can positively affect intestinal health and immune response [142], and may alleviate stress through the brain–gut axis [32]. In humans, GA has been reported to reduce the formation of free radicals and enhance innate immune activation [143], inhibit the production of ROS, nitric oxide, and the release of pro-inflammatory cytokines [144], and increase macrophage phagocytosis to improve immune regulation activity [145]. In addition, GA can induce a shift of intestinal microbial groups toward more favorable composition and promote the production of short-chain fatty acids (SCFAs) [146], which can serve as neuroactive substances further affecting the nervous and immune systems of the body [147]. Collectively, these activities of GA have positive significance for reducing the damage from oxidative stress. Yang et al. (2022) verified this in dogs by showing that GA markedly reduced diarrhea and caused a moderate decline of serum cortisol and heat shock protein (HSP) 70 levels in puppies after transportation [32]. The same study also reported that GA alleviated the oxidative stress and inflammatory response induced by transportation, and maintained the stability of intestinal flora and the content of short chain fatty acids [32]. In addition, the fecal and serum metabolomic analyses revealed that GA markedly reversed the abnormalities of nutrient metabolism caused by stress [32]. Tannic acid extracted from gallnut (a widely used traditional medicine in China) inhibited the secretion of serum stress hormones (i.e., COR, GC, and ACTH) and the expression of heat shock protein 70 to protect dogs from stress-induced oxidative damage and inflammatory response [106]. Dietary supplementation with pomegranate peel extract (PPE) had a positive impact on the antioxidant status in dogs, improving indices of erythrocytic antioxidants, namely, reducing glutathione, catalase, glutathione peroxidase, and glutathione S-transferase, together with a reduction in lipid peroxidation [148]. Resveratrol, a natural phytoalexin contained in wine, can reduce the level of ROS and MDA, improve the activities of SOD, GPX, and CAT activities, and improve the ratio of reduced glutamate to oxidized glutamate in cat models in which hepatotoxicity was induced [149]. Pinus taeda hydrolyzed lignin is a polyphenol mixture that can increase the activity of SOD, CAT, and GPx to improve antioxidant capacity in healthy dogs [107]. Curcumin extracted from curcuma longa can also enhance total antioxidant capacity by improving the activities of ROS, CAT, SOD, and GPx, and relieve inflammation by reducing lymphocytes and globulin level in dogs [108].

In vitro experiments with canine and feline cells have also revealed the antioxidative potential of some other plant or seed extracts. For example, quercetin is a natural occurring bioflavonoid that can increase GSH and decrease ROS in methimazole-induced oxidative stress in feline kidney epithelial cells [150]. Morin, also a flavonoid, can enhance the antioxidant capacity of hydrogen-peroxide-induced oxidative-stressed canine kidney cells by increasing the activities of SOD and CAT, and reduce mitochondrial oxidative damage and apoptosis [151]. Grape seed proanthocyanidin extract, alone or together with resveratrol, has also been proved to reduce ROS production in canine lens epithelial cells [152]. However, the antioxidative effects of some substances require further verification through in vivo studies.

#### 5.1.2. Vitamins

Vitamin C has a strong antioxidant capacity that can reduce the damage of free radicals to cells by actively removing superoxide and other ROS [112]. Decreased vitamin C levels have been detected in dogs with naturally occurring gastric dilatation–volvulus [153]. However, in dogs with chronic heart failure, the concentration of vitamin C increases, which is considered to be a compensatory increase induced by chronic oxidative stress [154]. One study on kidney transplant dogs showed that the activities of SOD, GPx, and CAT were increased after feeding vitamin C, indicating improved antioxidative capacity [111]. However, another study showed that when fed an adequate diet, additional vitamin C supplementation had no significant impact on the antioxidant capacity and immune function of healthy dogs [155]. Lipid-soluble vitamin E is a chain-breaking antioxidant that reacts with lipid oxygen or lipid peroxide free radicals [156]. A study on dogs showed that vitamin E can prevent the increase in plasma malondialdehyde caused by exercise, which indicates that vitamin E has a positive role in preventing lipid peroxidation [110]. When vitamin E and C, and beta-carotene, were fed together to cats with renal insufficiency, the concentration of serum 8-OHdG decreased, indicating alleviated DNA damage from oxidative stress [113]. Vitamin B plays an important role in the health of the central nervous system [157]. Some mixed foods rich in vitamin B, fish oil, and other antioxidants have been shown to improve the cognitive function of cats [158] and dogs [159]. However, the direct effect of vitamin B on stress in dogs and cats remains to be explored. Taken together, the addition of vitamins B, C, and E may have a positive effect on the antioxidative capacity and health of the nervous system in pets [6,113].

#### 5.1.3. Minerals

The antioxidant and anti-stress abilities of minerals have long been investigated and applied, especially in combination with vitamins [160,161]. Representative trace elements include Fe, Zn, Se, and Mn. The role of these elements has been widely verified in a variety of species [162,163]. Studies have also shown that some dog skin diseases may be related to oxidative stress and zinc deficiency [164]. Organic selenium can reduce blood malondialdehyde levels and improve the activities of glutathione peroxidase, superoxide dismutase, and catalase, thus enhancing the antioxidant capacity of dogs with induced renal calculi [114]. In hyperthyroid cats, radioiodine can reduce urinary isoprostane, the high level of which reflects renal oxidative stress [115].

#### 5.1.4. Polyunsaturated Fatty Acids

Polyunsaturated fatty acids (PUFAs) are fatty acids with more than one double bond in their backbone. Omega-3 PUFAs are among the most commonly used in dogs and cats. The antioxidative effect of PUFAs is achieved through either the component of cell membranes to decrease their sensitivity to free radicals, or boosting the endogenous antioxidative system (e.g., increasing cellular concentration of super oxide dismutase or gluthatione peroxidase) [165]. Feeding fish oil, which is rich in omega-3 PUFAs, to police dogs can promote the activities of GPx and CAT, and reduce the levels of blood glucose, and total and LDL cholesterol, indicating that fish oil can improve the antioxidant capacity and alleviate oxidative stress caused by strenuous exercise in dogs [117]. Additionally, due to its critical role in the development and function of the central nervous system, PUFAs are often included in brain protection formulae, which have been shown to improve the cognitive function and behavioral health of cats [158] and dogs [132,159,166]. In rodent models, it has been approved that omega-3 PUFAs can reduce anxiety-like behaviors and improve cognition in animals subjected to early life stress [167], possibly through the regulation of intestinal microbiota and the function of the brain–gut axis, including HPA [168]. Even though not verified in pet dogs and cats, we suggest similar mechanisms involved with the behavioral improvements and dietary intake of PUFAs exist as in rodents.

#### 5.1.5. Thiols

Thiols or mercaptans are a class of organic compounds with antioxidative capacity because their chemical structure contains a sulfhydryl group that is easily oxidized. The representative ones are N-Acetylcysteine and α-lipoic acid [169]. Studies on cats showed that feeding N-Acetylcysteine can increase the important cytosolic antioxidant, reduced glutathione, under the oxidative stress induced by onion powder [170]. N-Acetylcysteine can also protect liver tissue from the oxidative damage induced by acetaminophen in cats [171].

Lipoic acid is a small molecule of both animal and plant resources that contains two thiol groups that may be oxidized or reduced. Lipoic acid and its reduced form, dihydrolipoic acid, are powerful antioxidants with amphiphilic character [172]. They can easily quench radicals, chelate metals, interact with and regenerate other antioxidants, increase endogenous glutathione activity, and attenuate the release of free radicals and cytotoxic cytokines by regulating the second messenger nuclear factor κB [172]. The powerful antioxidant properties of α-lipoic acid make it helpful in the ancillary treatment of many human diseases, such as cardiovascular diseases and neurodegenerative diseases [172]. As summarized in a review study, supplementation of α-lipoic acid in appropriate doses (i.e., 1–5 mg/kg/day) can be beneficial in dogs, helping to reduce and delay lens opacities in diabetic dogs, reduce biomarkers of osteoarthritis, and when supplemented together with other antioxidants, reduce cognitive dysfunction and improve learning in senior dogs [173]. Even though α-lipoic acid can be safe and well tolerated by humans or animals, the recommendation of use in cats is rare because they are extremely sensitive to the toxic effect of α-lipoic acid compared to other species [174].

In humans and other animal species, additional substances and/or dietary formulae have been identified and investigated for their antioxidative function, such as saccharicterpenin, which is a new natural additive mainly extracted from Camellia plants [175], and the methionine/lysine proportion in the diet [176]. This evidence indicates that there are still many nutritional strategies with antioxidant potential that remain to be developed, and their application in relieving oxidative stress in cats and dogs requires further verification. Additionally, studies have suggested the use of a combination of different ingredients to achieve better antioxidative effects. For example, dietary supplementation of an antioxidant mixture containing quercetin (Q), resveratrol (R), curcumin, and vitamin E was shown to counteract both the oxidative stress and the related side effects elicited by methimazole treatment in hyperthyroid cats [177].

### 5.2. Anxiolytic Agents

#### 5.2.1. Gamma-Aminobutyric Acid and Its Receptor Agonists

Gamma-aminobutyric acid (GABA) is a small non-protein amino acid that is produced in the brain and other parts of the body (e.g., β cells of the pancreas, gastrointestinal tract, and endothelium) [178]. In the mammalian brain, GABA acts as the main inhibitory neurotransmitter and is widely known for its effect on anxiety- and stress-related disorders [179]. Peripheral administration of GABA is not effective in increasing its concentration in the brain due to the high polarity of the structure which limits its passage through the blood–brain barrier (BBB) [180]. Alternatively, many anxiolytic drugs/substances were developed to target GABA receptors [181]. Studies have shown that oral administration of alpha-casozepine, a milk-sourced lipophilic decapeptide that can cross the BBB and act on GABA receptors as an agonist, was effective in the management of anxiety disorders such as social phobias in domestic cats [121]. Alpha-casozepine was also shown to decrease the score of emotional disorder evaluation in dogs (EDED) [120] and reduce anxiety behavior and serum cortisol in anxious dogs [119]. The mixed addition of alpha-casozepine and L-tryptophan in diet reduced the urinary cortisol of cats [122] and reduced the inactive time of cats in unfamiliar environments [123]. The above evidence shows that alpha-casozepine, as a GABA receptor agonist, can alleviate anxiety and reduce stress in dogs and cats.

#### 5.2.2. L-Tryptophan

Tryptophan is the precursor for the synthesis of neurotransmitter 5-hydroxytryptamine/serotonin (5-HT), and the central serotonergic system is associated with fear- and anxiety-related states and stress responses [182]. An anxiolytic effect of a dietary addition of tryptophan is likely achieved by facilitating central 5-HT synthesis and signaling [58]. In cats, L-tryptophan is often tested together with alpha-casozepine [122]; therefore, the effect of tryptophan on stress management in cats requires further verification [123]. In dogs, the addition of tryptophan to the diet can increase the plasma tryptophan concentration and the ratio of tryptophan to large neutral amino acids [124]. Tryptophan will compete with large neutral amino acids (LNAAs) for transporters to cross the BBB [183]. When tryptophan and LNAAs were supplemented at a ratio of 0.075:1, the serum serotonin increased and the stool score improved in training dogs [125]. In addition, tryptophan supplementation can reduce attacks related to territorial domination [126] and stress-related anxiety behaviors [127] in dogs. Taken together, the dietary intake of tryptophan has the potential to alleviate anxiety in cats and dogs, but other factors that are involved in regulating tryptophan synthesis of 5-HT need to be considered, such as the ratio of tryptophan to LNAA, the alternative kynurenine pathway, and the activity of key enzymes including tryptophan hydroxylase [58].

#### 5.2.3. Theanine

Theanine, chemically named N-ethyl-L-glutamine, is an amino acid unique to green tea leaves that can compete with L-glutamic acid for the binding of glutamate receptors in the brain to exert its anti-stress effect [184]. Relevant studies in rat models have shown that theanine intake increases the concentration of 5-HT and dopamine in the brain [185]. In humans, theanine was shown to reduce the heart rate and relieve elevated blood pressure during stress, and weaken the stress response of the autonomic nervous system induced by physical and psychological stress [184]. In cats, theanine was shown to be effective in improving undesirable manifestations of stress, especially inappropriate elimination [130]. Theanine can also reduce the global anxiety scores in storm-sensitive dogs, as reflected by reduced anxious behaviors (e.g., drooling, following people, pacing, panting, and hiding) and latency to return to a baseline behavioral state after the storm ends [128]. In addition, a study suggests that theanine is effective for reducing fearful behavior toward unfamiliar human beings in dogs [129]. The above research shows that dietary administration of appropriate theanine may serve as a promising strategy for relieving stress and improving anxious behaviors in dogs and cats.

#### 5.2.4. Diet with Differed Macronutrient Composition

The composition of certain nutrients in the diet may also impact animal behavior. An earlier study revealed that incorporating more protein in the diet in exchange for an isoenergetic amount of fat resulted in a trend toward decreased dominance aggression but increased territorial (fear) aggression in dogs [186]. This change of dietary nutrients on behavior may be associated with tryptophan concentrations since diets with different protein contents (11.8, 16.9, and 22.2 g/MJ) are linearly correlated with their tryptophan levels (67, 105, and 115 mg/MJ). However, a ketogenic diet high in fat, but low in protein and carbohydrate content, was shown to reduce the attention-deficit/hyperactivity disorder and fear/anxiety of dogs with idiopathic epilepsy [131]. A possible mechanism is that the high content of medium chain triglyceride in the diet alters the energy metabolism in the brain which may contribute to behavioral changes [131]. More studies are required to determine the mechanisms underlying the connection between dietary nutrient composition and animal behaviors.

### 5.3. Probiotics and Prebiotics

Animal and human studies have shown that gut microbiota can be involved in the regulation of stress/emotion factors such as serotonin synthesis [187], brain-derived neurotrophic factor [188], and cortisol [189], thereby participating in the management of an individual’s stress level and related psychiatric symptoms. A microbial metabolite converted from tyrosine, 4-ethylphenyl sulfate (4-EPS) has recently been shown to contribute to the mechanism involving gut–brain interaction [190]. The metabolite can enter the brain, damage oligodendrocytes and reduce myelination of neuronal axons, thus inducing anxiety behavior [190]. The mixture of prebiotics, fish oil, and polyphenols can reduce the content of plasma 4-EPS and anxiety-related metabolites in dogs [136]. Meanwhile, the relative abundance of *Blautia*, *Bacterioides*, and *Odoribacter* was decreased, which are found to be decreasing in patients with anxiety [136]. A recent study in dogs reported that *S. boulardii* (1 × 10^9^ CFU di/kg of feed) reduced fecal calprotectin, IgA, and cortisol, indicating that *S. boulardii* may play an active role in alleviating intestinal inflammation and reducing stress hormone secretion [137]. Therefore, we can infer that improving the composition of intestinal flora may have therapeutic potential in relieving anxiety and stress. In terms of probiotics and prebiotics that can benefit gut health in pets, there are more studies in the literature that have provided evidence [135,138,139,191] but their direct effects on regulating stress and related behaviors are yet to be determined. However, probiotics and prebiotics have many positive effects on intestinal and neural health, which is expected to also play a role in relieving stress and related symptoms [192].

## 6. Conclusions

In the modern domestic environment, dogs and cats are regularly faced with various stress problems. Causes of stress include uncomfortable environments and conflicts in social life. When dogs and cats perceive stress, a series of physiological changes occur in the body, mainly mediated by the HPA and SAM axes. At the same time, oxidative stress has also been proved to be highly correlated with stress response. However, intestinal health is of great significance, especially in regulating dog and cat behaviors via the gut–brain axis. If stress is not alleviated, it may cause gastrointestinal diseases, urinary tract diseases, decreased immunity, abnormal behavior, and some cardiovascular problems. Dietary supplementation (e.g., antioxidants, anxiolytic agents, and probiotics) is conducive in alleviating the systemic changes associated with pet stress. Through this review, we provided insight into potential future research directions. Some small peptides and amino acids, such as alpha-casozepine and theanine, may act as agonists for receptors in the neuron system and thus show anxiolytic effects. Plant extracts (e.g., gallic acid and tannic acid) may have great potential in alleviating oxidative damage and promoting intestinal flora, which may be of great significance in improving intestinal stress symptoms. However, much remains unclear about how to apply the different dietary strategies into stress management (e.g., exact functions, side effects, and application guidelines). Overall, stress management and control in pets through dietary strategies is a systematic project that requires multifaceted efforts and sustained research.

## Figures and Tables

**Figure 1 antioxidants-12-00545-f001:**
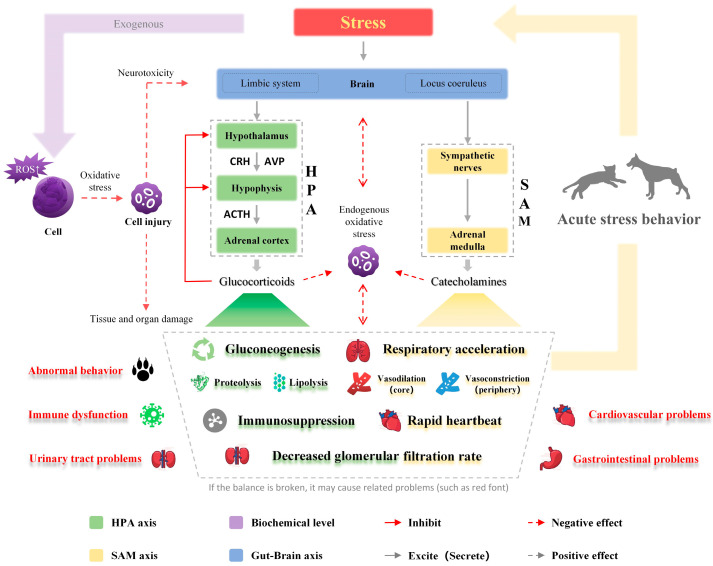
Regulatory mechanism of stress response. The components highlighted in green are mainly affected by the hypothalamus–pituitary–adrenal axis, and those in yellow are mainly affected by the sympathetic–adrenal medulla axis. It is worth noting that only the main impact is presented, and there is a broader and more complex relationship between the two systems. The elements in red indicate the possible harm of stress. Stress can induce oxidative stress to cause ubiquitous damage to cells, tissues, and organs. ROS, reactive oxygen species; CRH, corticotropin-releasing hormone; AVP, arginine vasopressin; ACTH, adrenocorticotropic hormone.

**Figure 2 antioxidants-12-00545-f002:**
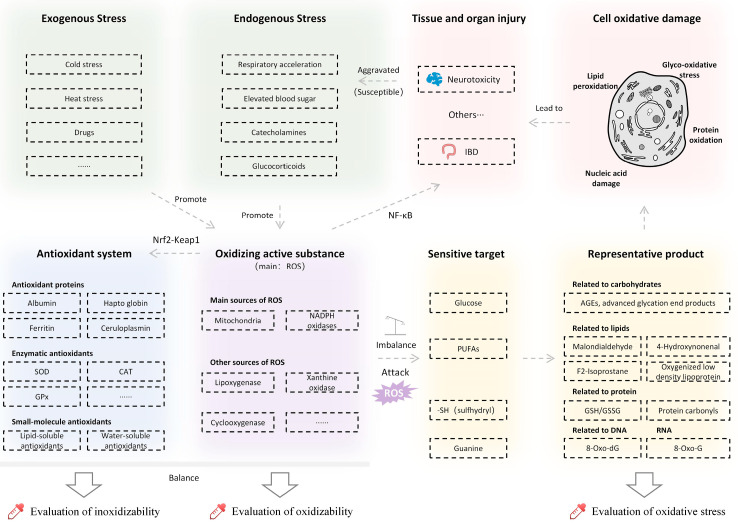
Mechanism and influence of oxidative stress. SOD, superoxide dismutase; CAT, catalase; GPx, glutathione peroxidase; 8-oxo-dG, 8-oxo-2’-deoxyguanosine; 8-oxo-G, 8-oxo-guanine; GSH, reduced glutathione; GSSG, oxidized glutathione disulfide; IBD, inflammatory bowel disease.

**Table 1 antioxidants-12-00545-t001:** Dietary strategies for stress alleviation in cats and dogs.

Active Ingredients	Resources	Mechanism	Species	Dosage	Measurements	Supportive/Negative	Reference
Gallic acid	Fruits, vegetables, and medicinal plants	Antioxidants; intestinal health	Dog	500 mg/kg	SOD and CAT ↑; TNF- α ↓; IL-1 β ↓; diarrhea rate ↓; SCFAs-producing bacteria ↑; serum cortisol and HSP70 ↓	Regulate intestinal flora to alleviate oxidative stress and inflammatory reaction	[32]
Tannic acid	Gallnut	Antioxidants; intestinal health	Dog	2.5 g/kg	Serum COR ↓; GC ↓; ACTH ↓; HSP70 ↓; beneficial bacteria ↑; pathogenic bacteria ↓; fecal butyrate ↑	Regulate intestinal flora to alleviate stress injury	[106]
*Pinus taeda* hydrolyzed lignin (PTHL)	*P. taeda* (tree)	Antioxidants	Dog	/	SOD, CAT, and GPx activity ↑	Antioxidation	[107]
Curcumin	*Curcuma longa*	Antioxidants	Dog	32.9 mg/kg	ROS ↓; CAT, SOD and GPx ↑; total antioxidant capacity ↑; lymphocytes and globulin levels ↓	Enhance antioxidant capacity and alleviate inflammatory reaction	[108]
A blend of essential oils and vitamin E	Essential oils (cloves, rosemary, and oregano)	Antioxidants	Dog	/	Non-protein self-sustaining group ↑; glutamate S-transfer ↑; ROS ↓	Antioxidation	[109]
Vitamin E	Commercial sources	Antioxidants	Dog	500 mg	Prevent the decrease in PON1 activity and EMF, and the increase in plasma MDA.	Alleviate oxidative stress	[110]
Vitamin C	Commercial sources	Antioxidants	Dog	/	SOD, GPx, and CAT ↑;	Antioxidation	[111]
			Dog/cat	/	ROS ↓; improve the blood flow distribution, promote the synthesis of catalamine and arginine vasopressin, regulate immunity, and inhibit the activity of cytotoxic T cells	Relieve the damage caused by oxidative stress and inflammatory reaction	[112]
VE and VC and beta-carotene	Commercial sources	Antioxidants	Cat	VE: 742 mg/kg; VC: 84 mg/kg; beta-carotene: 2.1 mg/kg	Serum 8-OHdG ↓	Reduce DNA oxidative damage	[113]
Selenium	Selenium yeast	Antioxidants	Dog	0.3 mg/kg	MDA ↓; GPx, SOD, and CAT ↑	Antioxidation	[114]
Radioiodine	Commercial sources	Antioxidants	Cat	/	Urinary free 8-isoprotenates ↓	Alleviate lipid peroxidation	[115]
*Saccharomyces cerevisiae* fermentation product	*S. cerevisiae* fermentation	Antioxidants	Dog	0.13%	Serum MDA and 8-isoprotenates ↑; the expression of blood COX-2 and MPO mRNA ↓	Inhibit innate immune activation to alleviate inflammation	[116]
Fish-oil-based foods	Commercial sources	Antioxidants	Dog	/	GPx and CAT activity ↑; blood glucose and total and LDL cholesterol ↓	Antioxidation and reduce blood sugar and blood lipid	[117]
Melatonin	Commercial sources	Antioxidants	Dog	0.3 mg/kg	Serum SOD, GPX, and CAT ↑; MDA ↓	Enhance antioxidant capacity to relieve oxidative damage	[118]
α-casozepine	A tryptic bovine αs1-casein hydrolysate	Anxiolytic agents	Dog	/	Anxiety behavior ↓; serum cortisol ↓	Relieve anxiety and improve behavior; reduce stress hormone secretion	[119]
			Dog	Closely 15 mg/kg BW	Score of emotional disorder evaluation in dogs ↓;	Relieve anxiety and improve behavior	[120]
			Cat	15 mg/kg BW	anxiety score ↓; different items (fear of strangers, contact with familiars, general fears, fear-related aggressions, and autonomic disorders) ↓	Relieve anxiety and improve behavior	[121]
α-casozepine and tryptophan	Commercial diet	Anxiolytic agents	Cat	α-casozepine: 15 mg/kg; tryptophan: 3.6 g/kg DM	The ratio of plasma tryptophan to large neutral amino acids ↑; urinary cortisol ↓	Promote tryptophan utilization and reduce stress hormone secretion	[122]
			Cat	/	The duration of cat inactivity decreases when placed in unfamiliar positions	Relieve anxiety and improve behavior	[123]
Tryptophan	Commercial sources	Anxiolytic agents	Dog	5.7 g/kg DM	Plasma Trp ↑; Trp/(large neutral amino acids) ↑	Promote tryptophan utilization; the impact on anxiety and behavior remains to be determined	[124]
			Dog	Trp: LNAA = 0.075:1	Serum serotonin ↑; improved stool scores	Relieve anxiety and reduce diarrhea	[125]
			Dog	Add extra 1.45 g/kg	Attacks related to territorial domination ↓	Reduce the stress of territorial competition	[126]
			Dog	/	Stress-related abnormal behavior ↓	Relieve anxiety and improve behavior	[127]
L-theanine	Commercial sources	Anxiolytic agents	Dog	50 mg (less than 10 kg), 100 mg (10–25 kg), 200 mg (more than 25 kg)/day	Anxiety scores ↓; drooling, following people, pacing, panting, and hiding ↓	Relieve anxiety and improve behavior	[128]
			Dog	50 mg (less than 10 kg), 100 mg (more than 10 kg)/day	Interactive behavior ↑	Relieve anxiety and improve behavior	[129]
			Cat	50 mg/day	Stress score ↓; inappropriate urination/defecation, fear-induced aggressiveness, hypervigilance/tenseness, or physical/functional manifestations of stress ↓	Relieve anxiety and improve behavior	[130]
Medium chain triglyceride diet	Commercial diet	Anxiolytic agents	Dog	5.5%	ADHD-related anxiety behavior ↓	Relieve anxiety and improve behavior	[131]
Medium chain triglyceride and Brain Protection Blend (BPB)	BPB including B vitamins, antioxidants, omega-3 fat acids, and arginine	Anxiolytic agents	Dog	6%/9%	Blood DHA, EPA, total omega-3 PUFAs, and omega-3/omega-6 ratio ↑; symptoms of cognitive dysfunction syndrome ↓	Promote brain health and improve behavior	[132]
Fish hydrolysate and melon juice concentrate	Commercial sources	Anxiolytic agents	Dog	F: 500 mg, M: 11 mg; double (BW more than 10 kg)	Interactive behavior ↑; stress behavior ↓	Relieve anxiety and improve behavior	[133]
Lemon balm, fish peptides, oligofructose, and L-tryptophan	Commercial diet	Anxiolytic agents	Cat	L: 0.1%; F: 0.1%; O: 0.5%; Trp: 0.08%	Average 24 h urinary cortisol/creatinine ratio ↓	Reduce stress hormone secretion	[134]
*Bacillus amyloliquefaciens* CECT 5940	Commercial bacteria	Intestinal Health	Dog	1 × 10^6^ CFU/g DM	The bacillus ↑; the coliforms ↓	Regulate intestinal flora	[135]
Polyphenols and omega-3 fatty acids	Fish oil and a polyphenol blend (citrus pulp, carrot, and spinach)	Intestinal Health; anxiolytic agents	Dog	/	Plasma 4-EPS ↓; anxiety-related metabolites ↓; Blautia, Parabacteroides, and Odoribacter ↑	Regulate intestinal flora to relieve anxiety	[136]
*S. boulardii *	Commercial bacteria	Intestinal Health	Dog	1 × 10^9^ CFU di/kg of feed	Fecal calprotectin ↓; IgA ↓; fecal cortisol ↓	Reduce intestinal inflammation and stress hormone secretion	[137]
A fiber–prebiotic–probiotic blend	Commercial sources	Intestinal Health	Dog	/	Fecal score ↓; blood lipid ↓; fecal IgA ↑	Enhance intestinal immunity and improve stool quality	[138]
*Enterococcus faecium* SF68	Commercial bacteria	Intestinal Health	Cat/Dog	2.1 × 10^9^ CFU/day	Diarrhea rate ↓	Reduce diarrhea	[139]

↑, increase; ↓, reduction; IgA, immunoglobulin A; NO, nitric oxide; COR, cortisol; GC, glucocorticoid; ACTH, adrenocorticotropic hormone; HSP70, heat shock protein 70; SOD, superoxide dismutase; CAT, catalase; GST, glutathione-S-transferase; GPx, glutathione peroxidase; ADHD, attention-deficit/hyperactivity disorder; ROS, reactive oxygen species; GSH, reduced glutathione; MDA, malondialdehyde; PON1, paraoxonase-1; EMF, erythrocyte membrane fluid; 4-EPS, 4-ethylphenyl sulfate; PUFA, polyunsaturated fatty acids; COX-2, cyclooxygenase-2; MPO, myeloperoxidase; LNAA, large neutral amino acids; Trp, tryptophan.

## Data Availability

The data presented in the study are available in the article.

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
