# Peer review of "Dietary Strategies for Relieving Stress in Pet Dogs and Cats"

_antioxidants, 2023, doi:10.3390/antiox12030545_

Round 1
Reviewer 1 Report
Overall, the is review manuscript is well prepared and concise and includes all the related info. I can confirm that the subject matter of this review paper is of interest and relevance for publication in Antioxidants.
Dear Authors, I read with your interesting manuscript. Some specific comments:
- add hypothesis and aim at the beginning
- the conclusion should not be a summary of discussion. Make sure the conclusion is short and solid. An idea may be to synthetize in 3-5 bullet the key results of the study, evidences and recommendation. This improvement will increase clearness and readability. Add a practical implications statement
- The manuscript should be prepared carefully according to the Journal's Guidelines
Evaluation of quality of the manuscript: 1. Originality of work (review) is good. 2. Methods of studies are adequate.
Author Response
Thank you very much for your suggestions. Your suggestions are very good and have brought great help to the article. In response to these suggestions, I have made modifications. Please see the attachment.

Reviewer 2 Report
Dear Authors
Congratulations on your work, I found it to be very well structured and well written and it summarises very well the advances in the treatment of the consequences of stress through dietary intervention.
I would like to note a few minor revisions:
Page 3, lines 86-87: You explain that the preganglionic neurotransmitter is acetylcholine, but you should also say that the postganglionic neurotransmitter is noradrenaline, which will induce the secretion of adrenaline into the bloodstream.
In this paragraph you mention the adrenal gland (medulla) and below you explain where it is located. Maybe you explain it here because this is the first time it comes up.
Page 12, line 391: When we talk about rodents, we are not talking about anxiety, but about anxiety-like behavior.
Page 14, line 187: blood-brain barrier = BBB (use the acronym)
Page 14, line 528: Rather than alleviating stress, it aimed to alleviate the systemic changes produced by stress.
Author Response

(The authors gave the same response as above.)
